# Production of Terretonin N and Butyrolactone I by Thermophilic *Aspergillus terreus* TM8 Promoted Apoptosis and Cell Death in Human Prostate and Ovarian Cancer Cells

**DOI:** 10.3390/molecules26092816

**Published:** 2021-05-10

**Authors:** Ayman A. Ghfar, Mohammad Magdy El-Metwally, Mohamed Shaaban, Sami A. Gabr, Nada S. Gabr, Marwa S. M. Diab, Ahmad Aqel, Mohamed A. Habila, Wahidah H. Al-Qahtani, Mohammad Y. Alfaifi, Serag Eldin I. Elbehairi, Bayan Ahmed AlJumah

**Affiliations:** 1Advanced Materials Research Chair, Chemistry Department, P.O. Box 2455, College of Science, King Saud University, Riyadh 11451, Saudi Arabia; aifseisi@ksu.edu.sa (A.A.); mahbila@ksu.edu.sa (M.A.H.); 2Botany and Microbiology Department, Faculty of Science, Damanhour University, Damanhour 22511, Egypt; mmmyco@gmail.com; 3National Research Centre, Chemistry of Natural Compounds Department, Division of Pharmaceutical Industries, El-Behoos St. 33, Dokki-Cairo 12622, Egypt; mshaaba@gmail.com; 4Department of Anatomy, Faculty of Medicine, Mansoura University, Mansoura 35516, Egypt; dr.samigabr@gmail.com; 5Department of Pediatric, Mansoura University Children Hospital, Faculty of Medicine, Mansoura University, Mansoura 35516, Egypt; nadasami23.ns@gmail.com; 6Molecular Drug Evaluation Department, National Organization for Drug Control & Research (NODCAR), Giza 12553, Egypt; marwa.db@gmail.com; 7Department of Food Sciences & Nutrition, College of Food & Agriculture Sciences, King Saud University, Riyadh 11451, Saudi Arabia; wahida@ksu.edu.sa; 8Biology Department, Faculty of Science, King Khalid University, P.O. Box 9004, Abha 61413, Saudi Arabia; alfaifi@kku.edu.sa; 9Cell Culture Lab, Egyptian Organization for Biological Products and Vaccines (VACSERA Holding Company), 51 Wezaret El-Zeraa St., Agouza, Giza 12311, Egypt; 10Department of Pharmaceutical Practice, College of Pharmacy, Princess Nourahbint Abdulrahman University, Riyadh 11451, Saudi Arabia; bayan.a.123456@gmail.com

**Keywords:** terretonin N, butyrolactone I, *Aspergillus terreus* TM8, anticancer activity, PC-3, SKOV3

## Abstract

The anticancer activity of terretonin N (**1**) and butyrolactone I (**2**), obtained from the thermophilic fungus *Aspergillus terreus* TM8, was intensively studied against prostate adenocarcinoma (PC-3) and ovary adenocarcinoma (SKOV3) human cell lines. According to this study, both compounds showed potent cytotoxicity towards ovarian adenocarcinoma cells (SKOV3) with IC_50_ 1.2 and 0.6 μg/mL, respectively. With respect to metastatic prostate cells (PC-3), the two compounds **1** and **2** showed a significantly promising cytotoxicity effect with IC_50_ of 7.4 and 4.5 μg/mL, respectively. The tested fungal metabolites showed higher rates of early and late apoptosis with little or no necrotic apoptotic pathway in all treated prostate adenocarcinoma (PC-3) and ovary adenocarcinoma (SKOV3) human cell lines, respectively. The results reported in this study confirmed the promising biological properties of terretonin N (1) and butyrolactone I (2) as anticancer agents via the induction of cellular apoptosis. However, further studies are needed to elucidate the molecular mechanism by which cellular apoptosis is induced in cancer cells.

## 1. Introduction

Despite the progress made in the field of cancer research, there is still a need to discover and develop anticancer therapeutic agents, representing the main target of numerous research groups worldwide, especially as cancer is responsible for millions of deaths across the whole world. The latter is responsible for approximately 7.6 million deaths worldwide and this is expected to increase to 13.1 million by 2030 [1]. For a long time, it was recognized that natural products represented the richest source of high chemical diversity, providing the basis for the identification of novel scaffold structures that serve as the starting points for rational drug design.

Structural diversity and complex molecular architectures represent key features of the natural products from fungi. They exhibit a remarkably wide range of biological activities, including anticancer properties [2,3,4,5,6,7,8]. Natural products identified and characterized from fungi, isolated from extreme habitats, are considered as promising lead compounds, e.g., in tumor therapy [9].

In the course of this work, two interesting compounds, terretonin N (1) [10] and butyrolactone I (2) [11], obtained recently from our investigation into thermophilic fungus *Aspergillus terreus* TM8 [12], were examined as anticancer agents against prostate adenocarcinoma (PC-3) and ovary adenocarcinoma (SKOV3) human cell lines. The chemical structures of the two compounds were fully identified and characterized in our recently published data [11,12,13]. In addition, our team reported the full taxonomy of the producing thermophilic *Aspergillus terreus* TM8 previously [10,11,12,13]. The structure of the isolated metabolites, shown in Figure 1, were assigned by comparison of their 1D and 2D NMR spectroscopy and ESI HR mass measurements with our recently published data [10,11,12,13].

## 2. Results

Production and structure assignments of terretonin N (**1**) and butyrolactone I (**2**) obtained from the thermophilic fungus *Aspergillus terreus* TM8 have been reported recently by us [10,11,12]. Details of the spectral data of both compounds (their physico-properties, NMR assignments (1D and 2D NMR), Tables of NMR data) are shown in the Appendix A.

### 2.1. Cytotoxicity Studies of Terretonin and Butyrolactone I

The cytotoxicity of terretonin N (**1**) and butyrolactone I (**2**) was evaluated by sulforhodamine B (SRB) assay towards PC-3 and SKOV3 cancer cell lines using serial concentrations (0.001 to 1000 μg/mL). The tested compounds (**1,2**) indicated analogous cytotoxicity profile against both tumor cells, where they showed the most potent cell-killing effect towards ovarian adenocarcinoma cells (SKOV3) at IC_50_ 1.2 and 0.6 μg/mL, respectively. With respect to metastatic prostate cells (PC-3), the two compounds (**1**, **2**) displayed a significantly promising cytotoxicity effect with IC_50_ 7.4 and 4.5 μg/mL, respectively (Table 1 and Figure 2).

### 2.2. Apoptosis Assaying

The isolated metabolites terretonin N (**1**) and butyrolactone I (**2**) were tested as apoptotic agents against the cancer cells of PC-3 and SKOV3. Both PC-3 and SKOV3 were treated with metabolites (**1**, **2**) for 48 h and stained with Acridine Orange/Ethidium Bromide (AO/EtBr), respectively. A higher percentage of early apoptosis was observed in human metastatic prostate cells (PC-3) compared to ovarian adenocarcinoma cells (SKOV3) which showed cells with late-phase apoptosis following treatment with terretonin N (**1**) and butyrolactone I (**2**), as shown in Figure 3 and Figure 4. In addition, the results showed that both terretonin N (**1**) and butyrolactone I (**2**) induced higher nuclear DNA fragmentation in SKOV3 than PC-3 cells (Figure 3 and Figure 4). In both PC-3 and SKOV3 cancer cells, the results showed that the activity of the metabolites terretonin N and butyrolactone I against cancer proceeded only via the apoptotic pathway (Figure 3 and Figure 4), whereas they had no necrotic effect against the progression of cancer cells.

## 3. Discussion

### 3.1. Cytotoxicity Studies of Terretonin and Butyrolactone I

The cytotoxicity of terretonin N (**1**) and butyrolactone I (**2**) was evaluated by SRB assay towards PC-3 and SKOV3 cancer cell lines using serial concentrations (0.001 to 1000 μg/mL). The tested compounds (**1**,**2**) indicated analogous cytotoxicity profiles against both tumor cells, where they showed the most potent cell killing effect towards ovarian adenocarcinoma cells (SKOV3) with IC_50_ 1.2 and 0.6 μg/mL, respectively. With respect to metastatic prostate cells (PC-3), compounds **1** and **2** displayed a significantly promising cytotoxicity effect with IC_50_ 7.4 and 4.5 μg/mL, respectively.

In a previous study conducted by us [9], terretonin N (**1**) announced no significantin vitro cytotoxicity towards human cervical cancer cells KB-3-1. Interestingly, in thisstudy, it has been shownthat terretonin (**1**) has a selective cytotoxic potency against PC-3 and SKOV3 cell lines. Alternatively, butyrolactone I (**2**) is a cyclin-dependent kinase (CDK) inhibitor [14], and Cdc2 and CDK2 kinase were found to be specific targets of butyrolactone I [15]. Butyrolactone I displayed antiproliferative activities towards the murine fibroblast L-929 and the leukemiaK-562 cell lines with IC_50_ values being 32.3 and 20.2 μM, respectively [16,17].

### 3.2. Apoptosis Assaying

The isolated metabolites terretonin N and butyrolactone I (**1**, **2**) were tested as apoptotic agents against PC-3 and SKOV3 cancer cells using AO/EtBr staining for 48 h. A higher percentage of early apoptosis was observed in human metastatic prostate cells (PC-3) compared to ovarian adenocarcinoma cells (SKOV3), showing cells with late-phase apoptosis after therapy with terretonin N and butyrolactone I. In addition, the results demonstrated that both terretonin N and butyrolactone I induced higher nuclear DNA fragmentation in SKOV3 cells than in PC-3 cells. In this investigation, the results showed that the activity of the metabolites terretonin N and butyrolactone I against SKOV3 cells, more than PC-3 cancer cells, only proceeded via an apoptotic pathway.

Research has been focused on the potential, valuable biological activity of terrestrial fungal metabolites, particularly cancer therapeutics [18,19]. The characterized fungal metabolites in this study surprisingly displayed remarkable antitumor properties via the induction of cellular apoptosis in both human metastatic prostate cells (PC-3) and ovarian adenocarcinoma cells (SKOV3) lines. Previously, some metabolites isolated from different fungi or lichens have progressed into anticancer activity [18,19,20], where anticancer activities displayed more potent cytotoxicity with more cellular apoptosis [18,19,20].

Fungi were shown to be a prospective source foundation of natural products with excellent pharmaceuticals. Metabolites isolated from fungi belong to a high class of diversity in chemical structures, namely aromatic compounds, amino acids, anthraquinones, butanolides, butenolides, cytochalasins, macrolides, naphthalenones, pyrones, and terpenes [21,22]. The cytotoxicity and related activities of the isolated fungal metabolites depend mainly on the diversity in their chemical structures and the functional groups/pharmacophores shown in them [23].

In our study, it was reported that the two secondary metabolites, terretonin N (**1**) and butyrolactone I (**2**) isolated from *Aspergillus terreus*TM8, could induce pro-apoptosis and late-phase apoptosis without the necrotic cell death of tumor cells. Conversely, other fungal metabolites activate various cell death pathways, apoptosis, autophagic cell death, and programmed necrosis (necroptosis) [23,24]. In a recent study, rasfonin (A304) isolated from *Talaromyces sp*., promoted autophagy and caspase-dependent apoptosis in cancer cells [19].

The results reported herein confirm the promising biological properties of terretonin N and butyrolactone I as anticancer agents against prostate (PC-3) and ovarian adenocarcinoma (SKOV3) cell lines by inducing cellular apoptosis. However, further studies are needed to illuminate the molecular mechanism, by which cellular apoptosis is induced in these cancer cells.

## 4. Material and Methods

### 4.1. General Experimental Procedure

The fungus *Aspergillus terreus* TM8 was previously isolated in Egypt from a sub-surface soil sample and its growth optimized at different temperatures (15–55 °C). The optimum growth of the fungus appeared at 45–50 °C and it was concluded that the strain has thermophilic properties [12,13]. In addition, the extraction, full characterization, and chemical nomenclatures of the isolated metabolites were fully addressed using different spectroscopic means of structural analysis, as previously reported [10,11].

### 4.2. Isolation and Identification of the Producing Fungus

Details of isolation and characterization of the extremophilic *Aspergillus terreus* TM8 have been reported in our recently published article [13].

### 4.3. Fermentation and Chromatographic Purification

Fermentation, working up of the obtained fungal extract followed by a series of chromatographic purifications including silica gel column, Sephadex LH-20, and preparative thin-layer chromatography afforded the desired two compounds, terretonin N (**1**, 300 mg) and butyrolactone I (**2**, 1.0 g) as the main products with colorless/crystalline appearance [10,11].

#### 4.3.1. Terretonin N (**1**)

The structure of terretonin (**1**) was identified by NMR (1D and 2D) and MS spectral data (see Appendix A) and by comparison with our published study [10].

#### 4.3.2. Butyrolactone I (**2**)

The structure of butyrolactone I (**2**) has been identified by NMR (1D and 2D) and MS spectral data (see Appendix A) and by comparison with our published articles [10,11].

### 4.4. Anticancer Activity

#### 4.4.1. Cell Culture

Prostate adenocarcinoma (PC-3) and ovary adenocarcinoma (SKOV3) human cell lines were purchased from the American Type Culture Collection (ATCC). Cells were cultured at the Roswell Park Memorial Institute (RPMI), 1640 Medium and 10% fetal bovine serum was added (FBS) (*v/v*) and incubated at 37 °C with 5% CO_2_ [25,26].

#### 4.4.2. Sulphorhodamine B (SRB)Assay

Cell viability of (PC-3 and SKOV3) human tumor cells after treatment with compounds **1** and **2** was tested by sulphorhodamine B assay (SRB). Cells were of healthy growth, cultured in a 96-well tissue culture plate for 24 h. The spent media were replaced with a new treated medium containing serial concentrations of each compound (0.01 to 1000 µg/mL), and cell control was included. Triplicate wells were then incubated for 72 h and later fixed with TCA (10% *w/v*) for one h at 4 °C. After washing three times in a dark environment, SRB dye 20 μL0.4% (*w/v*) was added for 10 min. Excess stain was removed using 1% glacial acetic acid (*v/v*), and the plates left to dry for 24h. SRB-stained cells were thawed with Tris-HCl and the color intensity was detected using a microplate reader at 540 nm. The IC_50_ values (dose of the treatment that reduces survival to 50%) were analyzed using SigmaPlot 12.0 software [27].

#### 4.4.3. Apoptotic Assay

DNA dyes acridine orange (AO) and ethidium bromide (EtBr) were used to test the morphological changes in cells treated (alive, apoptotic, and necrotic cells). Both AO and EtBr engaged with dead and alive cells that emitted green fluorescence when interpolated and hooked on DNA. EtBr was engaged by dead cells only, whereas it was omitted by alive cells and produced red, bloodshot fluorescence when intercalated into DNA. Healthy cells were plated on a sterile cover slide into a six-well plate. After 24 h incubation, the cells were treated with pre IC_50_s calculated for the chosen compounds and incubated for 48 h. Cells were washed with a cool PBS twice. Cells were marked with a combination of acridine orange 100μg/mL and ethidium bromide (AO/EB) 100μg/mL in PBS 1× on each well and then left for 5 min at RT. Cover slides were placed with cultured stained cells on slides and were observed immediately via a fluorescence microscope [28,29].

### 4.5. Statistical Analysis

Data are presented as mean SD unless otherwise indicated. Statistical significance was acceptable to a level of *p* < 0.05. All statistical analyses were performed using GraphPad InStat software, version 3.05 (GraphPad Software, La Jolla, CA, USA). Graphs were plotted using GraphPad Prism software, version 6.00 (GraphPad Software, La Jolla, CA, USA).

## 5. Conclusions

The results revealed that secondary metabolites terretonin N (**1**) and butyrolactone I (**2**) isolated from *Aspergillus terreus*TM8 have potential cytotoxicity towards both human metastatic prostate cells (PC-3) and ovarian adenocarcinoma cells (SKOV3) lines. The cytotoxic activity of the two fungal metabolites against cancer cells proceeded via the apoptotic pathway with little or no necrotic pathway. The data of the biological anticancer activity presented here, previous full characterization of these fungal metabolites, and future work in this direction will provide an important clue for potential drug development from fungal secondary metabolites in the treatment of human prostate and ovarian cancers.

## Figures and Tables

**Figure 1 molecules-26-02816-f001:**
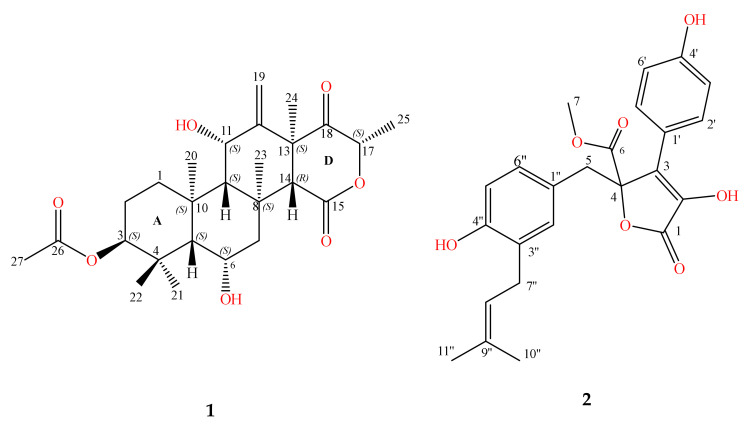
Chemical structures of terretonin N (**1**) and butyrolactone I (**2**).

**Figure 2 molecules-26-02816-f002:**
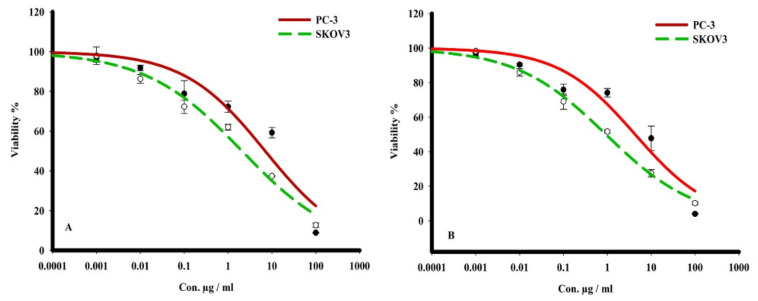
The dose response curves of the cytotoxicity of compounds **A** (**1**) and **B** (**2**) towards PC-3 and SKOV3 tumor cell lines, at different concentrations for 72 h. Cell viability was determined by SRB stain.

**Figure 3 molecules-26-02816-f003:**
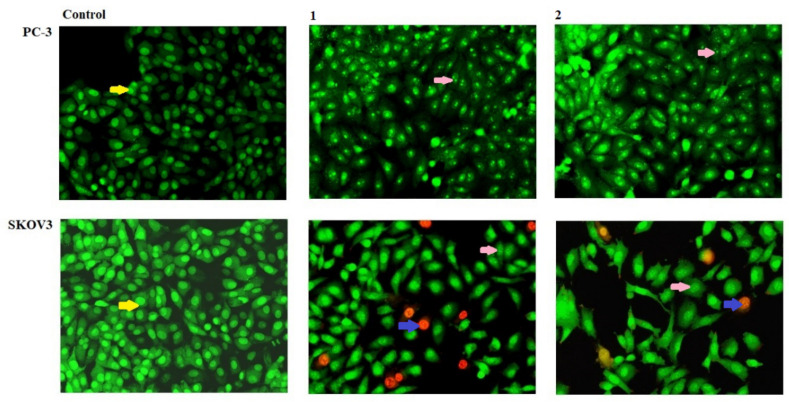
Morphological and nuclear changes using AO/EtBr staining that were evaluated by the effect of treatment with compounds **1** and **2** on apoptosis of PC-3 and SKOV3 human tumor cells after 48 h. The treatment-induced various nuclear changes (chromatin fragmented and condensation, nuclei condensation at 200×) in early and late apoptotic phases of the human tumor cells treated with compounds 1 and 2 respectively. Yellow arrows indicate live cells (cells with normal green nuclei). Pink arrows indicate early apoptotic cells (bright green nuclei with fragmented chromatin). Blue arrows indicate late apoptotic cells (orange-stained nuclei with chromatin condensation or fragmentation).

**Figure 4 molecules-26-02816-f004:**
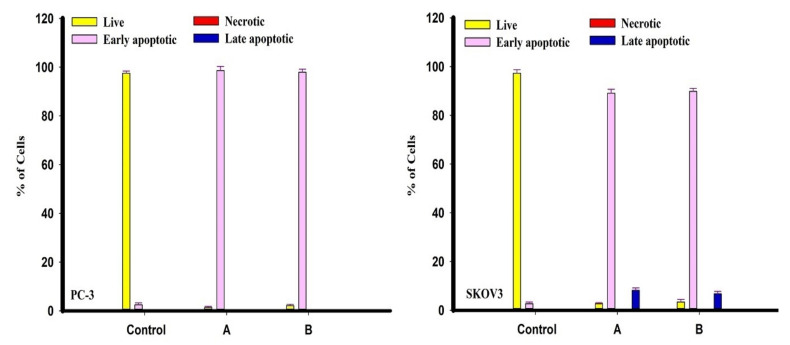
Percentage of apoptotic PC-3 (left) and SKOV3 (right) tumor cells after 48 h treatment with compounds A (**1**) and B (**2**) (mean ± SD of three independent experiments in three repeats each) compared to control cells.

**Table 1 molecules-26-02816-t001:** In vitro cytotoxicity of terretonin N (**1**) and butyrolactone I (**2**) against PC-3 and SKOV3 cell lines (IC_50_ [μgmL^−1^]).

Compound	Mwt (gmol^−1^)	IC_50_ [μgmL^−1^]	
		PC-3	SKOV3
Terretonin N (**1**)	462	7.4 ± 0.4	1.2 ± 0.2
Butyrolactone I (**2**)	424	4.5 ± 0.2	0.6 ± 0.1

## Data Availability

All data are available upon request from the authors.

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
