# Peer review of "Production of Terretonin N and Butyrolactone I by Thermophilic Aspergillus terreus TM8 Promoted Apoptosis and Cell Death in Human Prostate and Ovarian Cancer Cells"

_molecules, 2021, doi:10.3390/molecules26092816_

Round 1

Reviewer 1 Report

The manuscript entitled: “Synthesis of Terretonin N and Butyrolactone by Thermophilic Aspergillus terreus TM8 promoted apoptosis and cell death in human prostate and ovarian cancer cells” 
by Ayman A. Ghfar, Mohammad Magdy El-Metwally, Mohamed Shaaban, Sami A. Gabr, Nada S. Gabr, Marwa S. M. Diab, Ahmad Aqel, Mohamed A. Habila, Wahidah H. Al-Qahtani, Mohamed Y. Alfaifi, SeragEldin I. Elbehairi, and Bayan Ahmed AlJumah
reports a preliminar study on the anticancer activity against PC-3 and SKOV3 human cell lines of terretoninN and butyrolactone I, obtained from the thermophilic fungus Aspergillus terreusTM8, and already reported.
The paper is difficult to read mainly due to missing spaces between one word and another. 
Copies of NMR and Mass spectra are more suitable for a Supplementary material section instead of a section ‘results’ and, also in this case, they must be completed (e.g. Figure 5 must contain both chemical shift and integral values, in Figure 6 chemical shifts are omitted. 
In addition, some data (e.g. Table 1) have been already reported in a previous paper of authors. 
Figure 22 must be rescaled because some histograms are not well visible.
The concentration range of compounds administered to cells are not the same in discussion and experimental section.
The biological data are scarce,they might be improved performing other experiments.
In the whole, the manuscript must be re-written and deepened.

Reviewer 2 Report

This paper is a valuable contribution that explores the Running title: Synthesis of Terretonin N and Butyrolactone by Thermophilic Aspergillus terreus TM8 promoted apoptosis and cell death in human prostate and ovarian cancer cells; several techniques were explored and contains a great deal of information. However, I cannot recommend this manuscript to be published in this version in molecules due to the issues as follows.

The title of the manuscript is not correct because the authors does not the synthesis of the Terretonin N. it was isolated from the ethyl acetate extract of a solid-state fermented culture of Nocardiopsis sp.

In my opinion, the authors should move the entire characterization to supplementary material because the complete characterization has been published in different articles and leave only the anti-cancer activity in this manuscript.

Line 60. Thethermophilic fungus Aspergillus terreus TM8” should be “thermophilic fungus Aspergillus terreus TM8

The figure 12 is the same than in the publication molecules 2018,23,299

The full characterization of the Terretonin N was published in the Natural Product Research, 2018 vol. 32, No. 20, 2437–2446

The table 1 is the same that was published in molecules 2018, 23, 299

Please put in the same format all the spectroscopy data 1H, 13C one and two dimensions are in in different format.

Line 240. “The IC50” should be “IC50

Table 2. The NMR data of the carbons in position 9´´, 10´´ and 11´´ are missing

Table 3. The values of “IC50 mgmL-1” should be “IC50 mgmL-1

The references should be in format

Round 2

Reviewer 1 Report

The revised version has been improved, even if some typos are again present.

Reviewer 2 Report

This paper is a valuable contribution that explores the Running title: Production of Terretonin N and Butyrolactone I by Thermophilic Aspergillus terreus TM8 promoted apoptosis and cell death in human prostate and ovarian cancer cells; several techniques were explored and contains a great deal of information. However, I recommend this manuscript to be published in Molecules.
